# Weldability of 316L Parts Produced by Metal Additive Manufacturing

Hamdi Selmi [1,*], Jean Brousseau [1], Gabriel Caron-Guillemette [2], Stéphane Goulet [3], Jacques Desjardins [3] and Claude Belzile [4]

1   Département de Mathématiques, Informatique et Génie, Université du Québec à Rimouski, Rimouski, QC G5L 3A1, Canada
2   Alstom Transport Canada, La Pocatière, QC G0R 1Z0, Canada
3   Alstom Transport Canada, St-Bruno, QC J3V 6E6, Canada
4   Institut des Sciences de la Mer de Rimouski, Université du Québec à Rimouski, Rimouski, QC G5L 3A1, Canada
*   Correspondence: hamdi.selmi@uqar.ca

**Abstract:** The processes of metal additive manufacturing (AM) are no longer confined to rapid prototyping applications and are seeing increasing use in many fields for the production of tools and finished products. The ability to design parts with practically zero waste, high precision, complex geometry, and on-demand fabrication are among the advantages of this manufacturing approach. One of the drawbacks of this technique is the productivity rate, as the parts are made layer by layer, which also increases the production cost. Moreover, even the working space is limited, especially for the powder bed fusion technique. In view of these disadvantages and in order to guarantee the profitability of this process, it should be oriented to the production of complex components that have a limited volume with a design adapted to additive manufacturing. One solution with which to circumvent these drawbacks is to combine the 3D printing process with conventional manufacturing processes. When designing products, one may choose to use additive manufacturing to create locally complex parts and assemble them with parts produced by conventional processes. On the other hand, and due to the limited AM printing chamber space, it may be necessary to print large parts in multiple smaller parts and then assemble them. In order to investigate the weldability of stainless steel 316L parts produced by laser powder bed fusion (L-PBF), the mechanical behavior of different welding assemblies is tested. Five configurations are studied: non-welded AM specimens, two AM parts welded together, one AM part and one laser cut part welded together, two laser-cut parts welded together, and non-welded laser cut specimens. Welding is performed using the Pulsed Gas Metal Arc Welding process (GMAW-P). Specimen strength is assessed through static and fatigue tests. The results demonstrate that 316L AM parts are weldable, and the tensile and fatigue properties of L-PBF 316L welded components and welded laser cut components are comparable. GMAW-P welding led to lower fatigue results for AM components than for other configurations, but the difference is not important. It was observed that welding defects may have a direct impact on mechanical properties.

**Keywords:** additive manufacturing; gas metal arc welding; mechanical proprieties; stainless steel

## 1. Introduction

The contribution of additive manufacturing (AM) to the practice of creating commercial products has progressed enormously over the last 30 years. ASTM International now defines additive manufacturing as a process of merging materials, usually layer upon layer, to create objects from 3D digital data models [1]. AM is a promising technique that offers the possibility of creating fully functional parts in one operation without wasting much raw material. A major benefit of this technology is the elimination of constraints related to DFM (Design for Manufacturing) [2]. In relation to the Integrated Product and Process Development approach, two AM design strategies were suggested by Klahn et al. [3]:

"manufacturing-driven design strategy" and "function-driven design strategy". The first one remains faithful to the conventional design approach; however, the second one proposes an improved design and increases the functionality of the product. In addition to enhancing part design, the development of this technology has resulted in the production of a variety of materials, including metals, ceramics, polymers, and composites with mechanical properties (static and dynamic) comparable and sometimes improved compared to conventional manufacturing methods [4,5].

Metal applications in AM technology are becoming increasingly popular. The literature presents four developed techniques with which to fabricate metal components: direct energy deposition, powder bed fusion, sheet lamination, binder jetting, and cold spray deposition [6,7]. Due to their higher technology readiness levels (TRL) and high-density rate (near full density) [8], direct energy deposition (DED) and powder fusion are the two dominant AM technology families. The first consists of spraying the metal fed in powder or wire form on a laser source to form the part. One of its important advantages is that the DED effector can be fixed in a multiaxial machine, which allows for the building of parts without the need for support structures [2,9,10]. The second family is powder bed fusion (PBF), among which we find Laser Powder Bed Fusion (L-PBF), and Electron Beam Melting (EBM). These processes are capable of manufacturing high-quality and complicated geometry metal parts based on a wide variety of metals and alloys, such as carbon steel, stainless steel (316L,17-4 PH), aluminum (AlSi10Mg, Al2139), and titanium (Ti64, TiGr5). The PBF process is based on the creation of layers (sections) 20 to 150 μm thick. The energy source (laser or electron beam) fuses the powder of each superimposed 2D profile to build the complete part. Several experimental studies have focused on the mechanical properties of the parts produced by these technologies. The results show that the properties of additive manufacturing parts are comparable or even better than those manufactured using conventional processes such as cold rolling, casting, laser cutting and forging [11].

However, the PBF process has some drawbacks: the deposition rate of the process is low because printers build parts one layer at a time, the price of L-PBF machines is high because it is a new process, the print volume limits the overall dimensions of the printed elements, and the AM process requires parts to be supported during printing, which increases the post-processing time. Despite these disadvantages, PBF printing is gaining ground among all manufacturing processes. In order to guarantee the technical contribution and economic profitability of AM, the process should be used mainly for parts with high geometric complexity and limited dimensions. It is often observed that additive manufacturing allows the number of parts in an assembly to be reduced by increasing the complexity and functionality of products, which minimizes the number of assembly operations required. Ana et al. [12] present the industrial applications of metal AM, which are based mostly on automotive, aerospace, and industrial machines. They demonstrate that this technology is mainly used to enrich subassemblies with highly complex parts that can be joined mostly by mechanical solutions such as bolting and clamping. In addition, geometrically complex parts that are too large to be printed can be subdivided and assembled. AM is therefore a good mean to save manufacturing costs and improve the functionality of an assembly when it is used for the best it has to offer.

Improving the functionality of an assembly by integrating AM parts may save on manufacturing costs. When integrating AM parts into a new product, different means of assembly may be used, including welding. However, welding is very rarely used as a means of joining AM parts, perhaps due to a lack of studies and research on the subject [13–15]. To determine the eligibility and reliability of this approach, it becomes necessary to study the possibility of joining by welding two AM parts or one AM part with one CM (Conventional Manufacturing) part. In principle, since 3D printing is a process similar to welding, most materials used in additive manufacturing should be weldable. Different research projects have been undertaken using materials such as stainless steel, aluminum, and titanium [14,16,17]. Liu et al. [18] also present the idea of repairing the process of aerospace

parts through the welding of AM. For some applications, the cost of some parts and sub-assemblies is high. Rather than replacing them, repair using welding is a solution that can increase service life. For example, a cracked additive manufacturing part could be repaired to lengthen its life span. To increase the use of additive manufacturing, process and assembly methods must become better known, easier to use and more productive.

Experiments with laser welding and 316L stainless steel have shown the ease of having full weld penetration of AM parts compared to CM parts [13]. The authors pointed out that the much rougher and matte surface of AM parts induces a higher absorption rate and explains the full penetration that can be created with less laser power compared to rolled or machined parts. However, there is very little information on the welding of additively manufactured stainless steel parts. In another research based on a comparative study between laser-welded CM and PBF components made of 316L stainless steel, Mokhtari et al. [14] observed a significant hardness variation in the heat-affected zone of AM welded parts compared to conventional CM welded parts. The authors investigated the mechanical behavior by conducting tensile tests. For all samples, failure occurred in the fusion zone. The test specimens showed an elastoplastic behavior and large ductility. In order to inspect the welding quality of AM components, Andrea et al. [19] evaluated the microstructure of AM parts made of Ti6Al4V titanium and AlSi10Mg aluminum welded, respectively, by Linear Friction Welding (LFW) and Friction Stir Welding (FSW). In both cases, the observed joint was free of defects. The authors interpreted that friction welding does not significantly affect mechanical properties such as hardness, although it does cause a microstructure transformation. The weldability via gas tungsten arc welding GTAW of 316L PBF components welded to a conventional 316L wrought component was investigated by Huysmans et al. [15]. Without any post-processing, the results obtained confirm that the welded parts have satisfactory tensile mechanical properties. The specimens fractured in the conventional metal. Heterogonous GTAW welding was also studied by Laitinen et al. [20]. The authors noted that joining cold rolled and AM 316L parts provides good-quality welds with a better ultimate tensile strength than the AM base metal. Depending on welding speed, the rupture occurred in the welded metal at a welding speed of 1000 mm/min and in PBF metal at higher speeds.

In the various conducted studies, the process used to manufacture the specimens was mainly powder bed fusion [13–15,20], while the scanning parameters (power and laser speed) were maintained according to the recipes recommended by the suppliers. The relationship between construction parameters, microstructures, and mechanical properties were studied [21–23]. In order to assess weldability, different aspects were observed, including tensile, corrosion, hardness, and microstructure tests. Despite the importance of fatigue resistance, there are no studies on the subject of fatigue life prediction of welded AM parts with heterogeneous parts.

316L stainless steel (UNS S31603) is an austenitic steel (cubic face centered structure) used for many applications requiring high corrosion resistance and high mechanical properties. Compared to carbon steels (AISI1010, AISI1018...) and alloy steels (AI-SI4130, AISI8620...) this steel contains a large amount of Cr and Ni. The Ni content stabilizes the austenitic phase at low temperatures and improves corrosion resistance [14]. The 316L steel is also known for its low carbon content, which is preferable when welding these materials, as it reduces intergranular corrosion of welds and heat-affected zones caused by the precipitation of carbides [24]. In powder form, 316L is widely used for additive manufacturing because of its low thermal conductivity, low sensitivity to oxygen, and high absorption at infrared wavelengths [25]. 316L is used in several AM processes, such as DED, L-PBF, and cold spray [15,26,27].

Additive manufacturing is a highly digital process; its combination with laser welding is interesting, as the two processes are similar in terms of the precision and fusion method. However, for some applications, such as field repairs or working in crowded spaces, manual welding may be more practical. Indeed, AM is more oriented to a limited number

of products, so the combination with GMAW-P (Pulsed Gas Metal Arc Welding Process) is relevant. Welding also enables several AM parts to be combined to form a larger assembly.

The aim of this study is to investigate the weldability of L-PBF AM parts made of 316L stainless steel through the manual welding technique GMAW-P. In order to investigate the weldability of 316L stainless steel parts produced via laser powder bed fusion (LPBF); the mechanical properties of different welding assemblies are tested. Five configurations are studied: a non-welded AM specimen, AM parts welded together, one AM part welded with one laser cut rolled part (AM-CM), laser cut rolled parts welded together (CM-CM), and non-welded laser cut rolled specimens (CM). The five configurations are compared by performing fatigue and tensile tests, as well as establishing the microhardness profile and assessing the weld quality through micrographic and macrographic observations.

## 2. Materials and Methods

The research project consists of testing welded and non-welded 316L specimens. The non-welded specimens are 3D printed and laser cut. The welded specimens are made from three configurations: AM-AM, CM-CM, and AM-CM. Therefore, a test plan targeting the mechanical properties of the different groups of specimens is conducted.

### 2.1. Test Piece Fabrication

For this study, EOS 316L powder is used. At the beginning of the project, the dispenser is filled with a batch of fresh powder. The powder is then recycled from one print to the other without adding fresh powder. Between each print, the used powder is sieved with an 80 μm sieve and mixed with the remaining powder in the dispenser. Figure 1 shows images of the powder taken with a scanning electron microscope (SEM) (SNE 4500M, SEC Co., Ltd., Suwon, Republic of Korea). The size of most of the powder grain size is between 30 μm and 40 μm, as shown, and some particles are merged. A Mastersizer 3000 (Malvern Panalytical, Malvern, UK) particle size analyzer is used to determine the particle size distribution of the powder. Three measurements are taken and the d-values D10, D50, and D90 of the particle size distribution are, respectively, 18.5 μm, 35.6 μm, and 65.3 μm (see Figure 2), where D10, D50, and D90 give the maximal particle size diameter that includes 10%, 50%, and 90% of the particles (volume weighted basis), respectively.

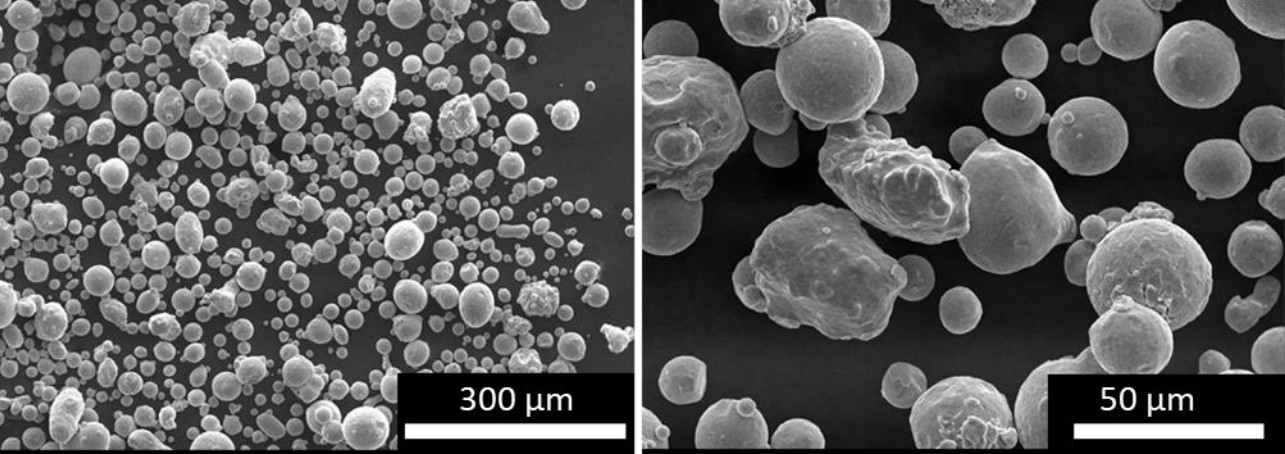

**Figure 1.** EOS 316L powder observed with SEM.

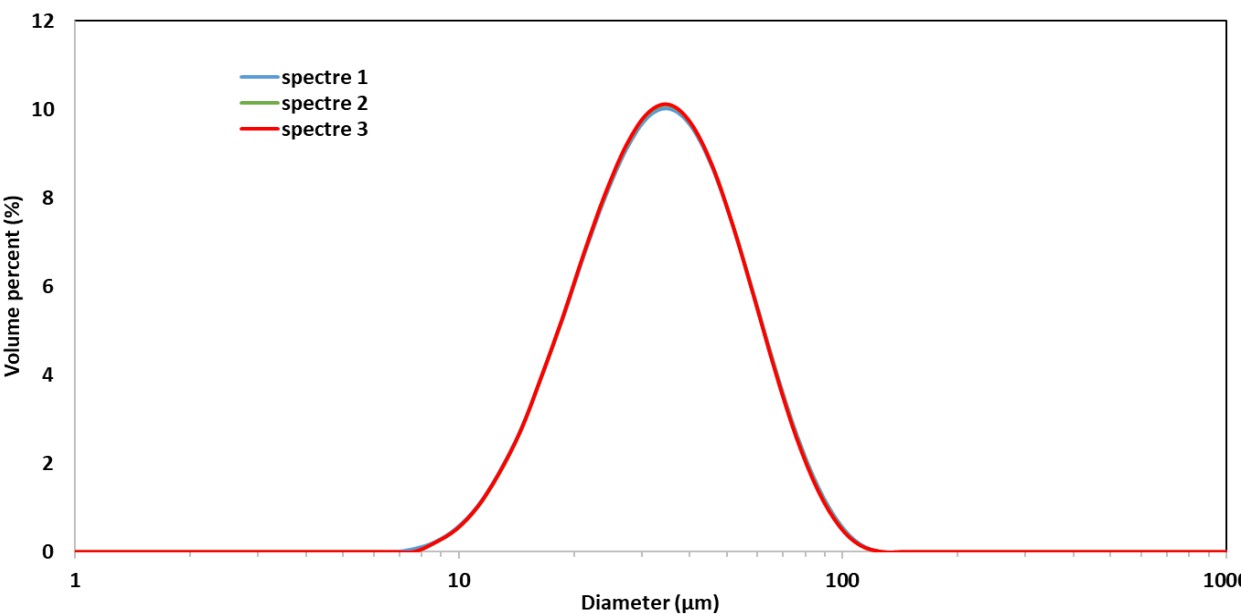

**Figure 2.** EOS 316L powder particle size distribution. The spectra from three measurements are practically overlapping.

According to EOS, the chemical composition of this powder is given in Table 1. Although the mechanical properties of 316L stainless steel printed parts have already been investigated [28], this study repeats the tests in order for comparison with literature and welded configurations.

**Table 1.** Chemical composition of EOS 316L powder.

| Fe | Cr% | Ni% | Mo% | C% | N% |
|---|---|---|---|---|---|
| Balance | 18.00 | 14.00 | 2.5 | ≤0.03 | ≤0.10 |

The additive manufactured full specimens and half specimens are made with the EOS M290 machine based on L-PBF technology. The manufacturing of the specimens is executed in three steps. The samples are constructed on a steel build plate and manufactured in a horizontal orientation (0°) relative to the build plate. The supports used for printing the specimens are solid and 4 mm high. The printing process takes place under Argon to protect the melt from oxidation and the maximum oxygen content is under 1.3%.

Table 2 presents the printing parameters used to manufacture the AM specimens, as defined by EOS.

**Table 2.** EOS 316L manufacturing parameters.

| Laser Power [W] | Laser Velocity [mm/s] | Layer Thickness [μm] | Hatch Distance [mm] |
|---|---|---|---|
| 195 | 1083 | 20 | 0.09 |

Figure 3 shows the different stages of the additive manufacturing process of specimens from digital preparation with EOSPRINT to surface machining post-treatment via the printing process and part cutting. Both the top and bottom sides are machined to have a similar surface finish (see Figure 3). In this study, because the specimens will be welded, no heat treatment is applied. The powder supplier mentions that heat treatment is optional and proposes stress relief and solution annealing. With the same manufacturing conditions and without any heat treatment, the mechanical properties measured by EOS are shown in Table 3.

**Table 3.** 316L AM mechanical proprieties measured by the supplier (horizontal mode).

| Yield Stress (MPa) | Ultimate Tensile Strength (MPa) | Elongation at Break (%) |
|---|---|---|
| 540 | 640 | 40 |

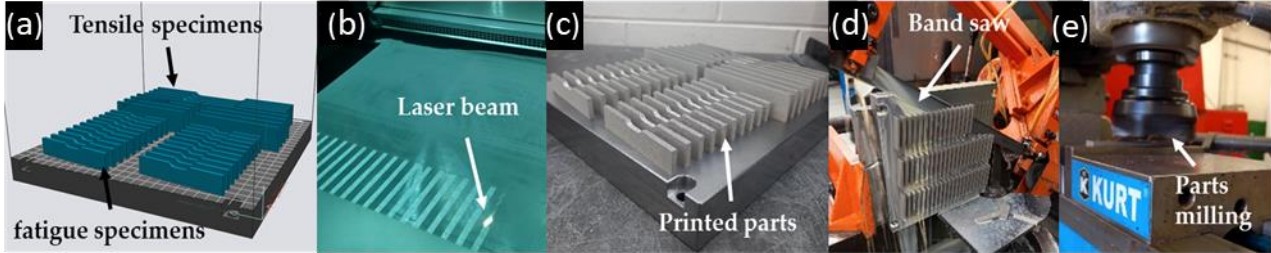

**Figure 3.** Additive manufacturing specimen preparation, (**a**) STL file preparation with EOSPRINT, (**b**) parts building, (**c**) finished print, (**d**) parts cutting, (**e**) supported surface machining.

Conventional specimens (half and full specimens) were provided by a local laser cutting and metal processing company. Table 4 shows the elementary composition of a conventional 316L sheet of metal according to the manufacturer mill test report.

**Table 4.** Chemical composition of base material for CM parts and specimens.

| Fe | Cr% | Ni% | Mo% | C% | N% | Mn% | Cu% | P% |
|---|---|---|---|---|---|---|---|---|
| Balance | 16.58 | 10.03 | 2.01 | 0.01 | 0.05 | 1.17 | 0.51 | 0.02 |

### 2.2. Test Piece Welding

Welding was performed by GMAW-P. This welding process was based on the recommendation of the industrial partner. Although laser welding is more efficient, manual GMAW-P welding is widely used in manufacturing industries and it represents the most probable technique for field repairs situations. The welding parameters are shown in Table 5.

**Table 5.** Welding operation mode.

| Filler Metal | Gas Flow | Gas | Backing | Transfer Mode |
|---|---|---|---|---|
| EN ISO 14343-A g 18 8 MN | 30–45 cfh | 98% Ar + 2% $CO_2$ | Copper | Pulsed |

| Current | Voltage | Advance speed | Filler wire diameter |
|---|---|---|---|
| DC (el.+) | 20 V | 320 in/min | 1 mm |

The filler metal used is EN ISO 14343-A g 18 8 MN, manufactured by Lincoln Electric. Its mechanical properties are as follows: Yield Strength 414 MPa; Tensile Strength = 607 MPa; Elongation = 37%.

To be more efficient, it was proposed to weld many specimens at a time; for this reason, the half AM and CM pieces were designed as rectangular shapes. This choice ensures that the specimens, stacked side by side, are in contact during welding and minimizes the welding path discontinuity. It also provides a means to minimize the number of weld imperfections that may occur at the start and stop of weld segments.

The specimens used to evaluate the AM-AM, CM-CM and AM-CM configurations are butt welded and the final geometry is designed according to the ISO 9692-1:2013 standard.

The specimens are welded at the factory of our industrial partner (Alstom Transport Canada, La Pocatière, QC, Canada). Each welding batch contains parts of the same configuration. For each configuration, a total of 3 tensile and 20 fatigue specimens are prepared. Based on the welding operation mode and in order to eliminate welding defects and ensure

complete penetration, the specimens are supported by a copper plate with a gap in the middle, which increases the melt volume. The spacing between the pieces (root opening) is equal to 4 mm (see Figure 4).

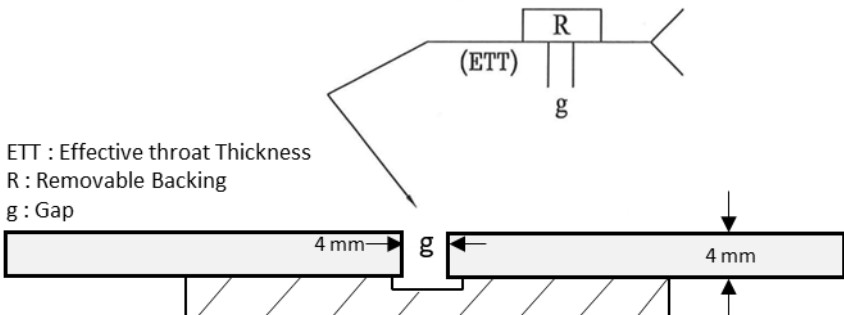

**Figure 4.** GMAW-P welding drawing.

Usually, the weld bead presents some anomalies at the beginning and at the end. In order to ensure repeatable welding results for the test specimens, stainless steel pieces of the same thickness as the test specimens are inserted at the beginning and at the end of each welding batch to provide run-on and run-off tabs at the weld ends, as suggested per good industrial practice. The jig contains clamps to fix the half specimens, avoid slight movements during welding, and reduce the angular distortion.

Before welding the test specimens, some samples are welded in order to confirm the welding procedure, evaluate the quality of welding, and measure distortion of the welded specimens. Figure 5 shows the assembly used to weld these specimens. According to a visual inspection, the preliminary results do not show any defects such as lack of penetration and cracks. The used welding procedure meets the requirements and offers results in accordance with the AWS D15.1 standard. After welding, specimens are separated from each other using a cutter wheel tool.

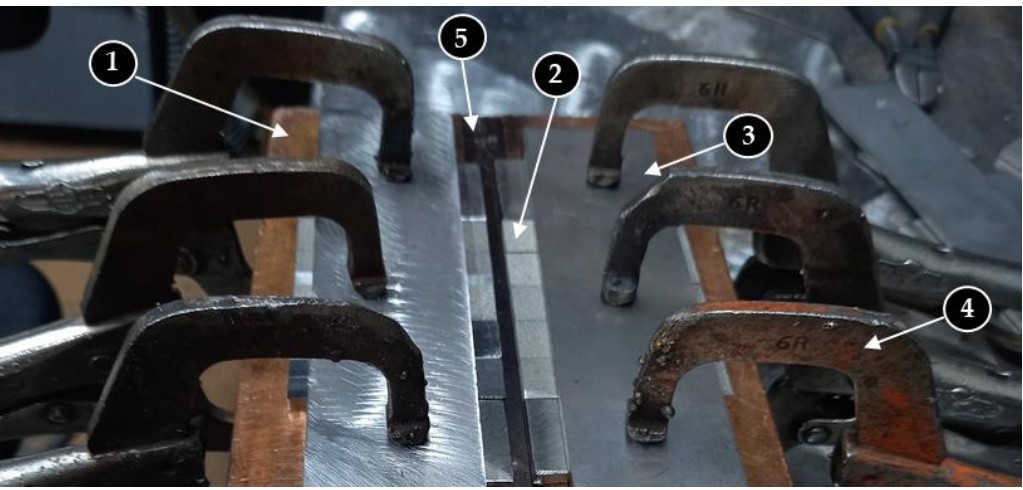

**Figure 5.** Welding jig: (1) Copper plate, (2) specimens welding side, (3) clamping part, (4) clamps, (5) groove.

*2.3. Angular Distortion Measurement*

Since welding is performed from one side, and as a result of the heat imposed during welding, an angular deformation is expected. Since the top side receives more heat, it becomes concave and the back side becomes convex. Sung-Wook Kang [29] studied the effects of angular distortion on stress concentration and fatigue life. It has been demonstrated that fatigue cracks appear on the welding side (the surface where the welding is applied). In

order to quantify the impact of this phenomenon on mechanical tests, angular distortion is measured using a dial indicator.

The measurements are taken on a batch of three specimens. The elevation is measured at both ends and in the centre. The values are converted into an angular deformation which varies from 0.12° to 0.37° for the fatigue specimens and from 1.21° to 1.65° for tensile specimens (see Figure 6). As this deformation is similar for all configurations, it is not considered as part of the study.

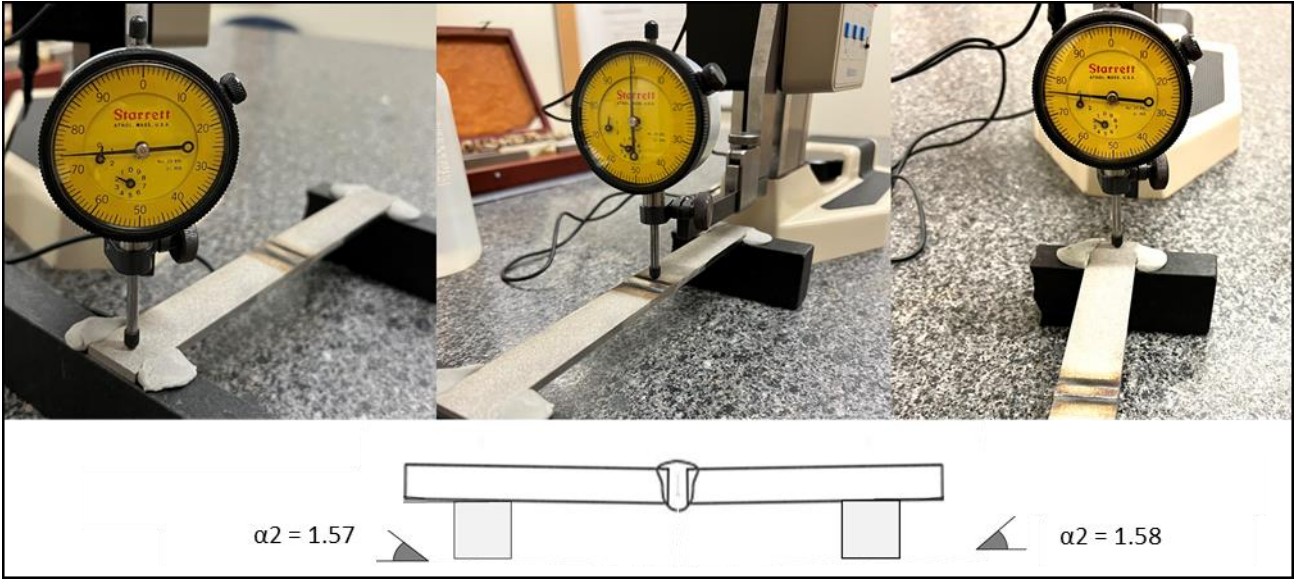

**Figure 6.** Distortion measurement example of AM-AM traction sample after welding.

### 2.4. Tensile Test Protocol

According to the ASTM E8/E8M, tensile testing is performed via milling on standard rectangular samples. This choice ensures ease of printing and material support removal. Specimen dimensions in mm post-treatment are shown in Figure 7. To assess the repeatability of the test, four tensile specimens are prepared for each configuration. The tensile tests are performed on an MTS 809 tensile testing machine; the test speed is set to 0.025 mm/s and the pressure applied by the clamping dies is equal to 2000 MPa in order to avoid any slippage.

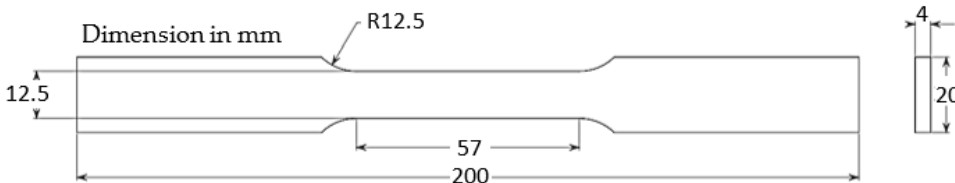

**Figure 7.** Geometry of samples for tensile test.

### 2.5. Fatigue Test Protocol

Axial fatigue test specimens are prepared by milling in accordance with the ASTM E466-15 standard. A sketch of the fatigue specimens highlighting their dimensions in mm is shown in Figure 8.

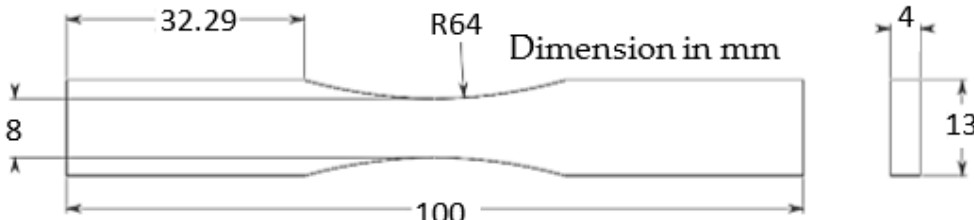

**Figure 8.** Geometry of samples for fatigue test.

The tests are performed at 20 Hz under a stress ratio of R= $\sigma_{min}/\sigma_{max}$ = 0.5. Two million cycles are considered a runout. The minimum and maximum stresses are calculated using the nominal stress formula $\sigma = F/A$ where F is the minimum or maximum applied force and A is the smallest cross sectional area of the specimen. Fatigue results are presented as a function of the alternating stress, $(\sigma_{max} - \sigma_{min})/2$.

The S-N curve is divided into two regions: finite and infinite life. For the finite portion of the curve, 9 samples are used and 3 load levels are considered for each configuration. Applied loads are established as a fraction of the ultimate tensile strength, (see Table 6). With the endurance limit at 2 million cycles, we use the staircase method based on Dixon and Mood method and as defined by ISO 12107: 2003 and test 9 samples for each configuration. The alternate stress increases or decreases with a step size equal to 10 MPa depending on whether the specimen survives or breaks. As suggested in the NF A03-405 application guide, the step of increase or decrease corresponds to 10 MPa for welded samples. The estimates of mean fatigue strength at 2 million cycles and its standard deviation for the different configurations are based on the formula presented by Snyder et al. [30].

**Table 6.** Different stresses applied for the determination of the S-N curve.

|  | Level | CM | AM | CM-CM | AM-AM | CM-AM |
|---|---|---|---|---|---|---|
| Applied | 1 | 155.58 | 148.61 | 142.32 | 140.96 | 141.63 |
| alternate | 2 | 135.11 | 140.99 | 135.06 | 129.27 | 130.18 |
| stresses (MPa) | 3 | 126.10 | 130.92 | 126.05 | 113.24 | 114.04 |

Table 6 presents the different levels of the alternating stresses applied in the finite life region for the 5 configurations.

### 2.6. Micrographic Observation and Microhardness Test Protocol

The microhardness measurements are performed with a CLEMEX automated tester. The samples are cut, coated, and polished to avoid the effect of surface roughness (Figure 9). The scale used is a Rockwell C (HRC) with a diamond indenter. The measurement method follows the ASTM E18 standard. The profiles are taken on a cross-section (perpendicular to the welding direction).

For micrographic observation, the specimens used for microhardness measurements are repolished. For cross-section observations, the surfaces are etched with "Kalling's reagent #2", which is the solution mentioned in the supplier's documents (EOS). A NanoImage SNE 4500M scanning electron microscope (SEM) is used for micrographic observations.

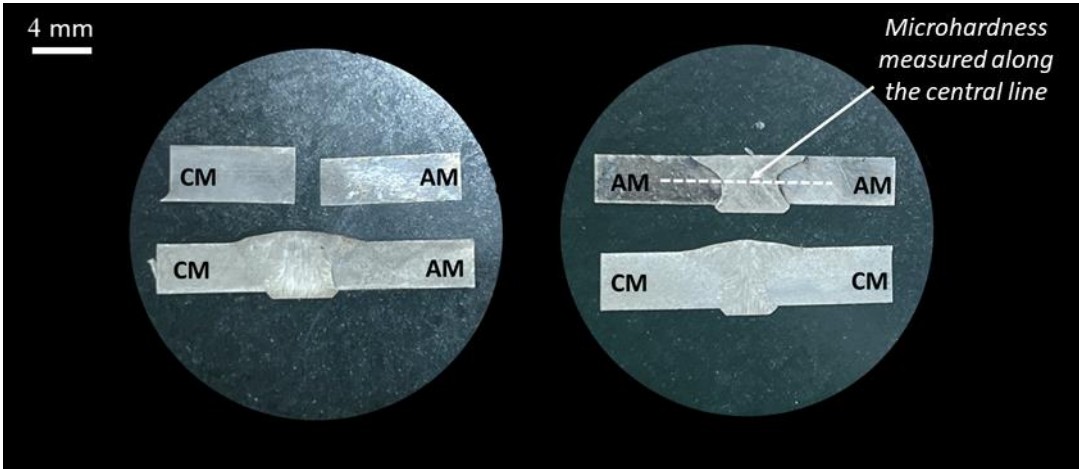

**Figure 9.** Microhardness and micrography samples.

## 3. Results and Discussion

### 3.1. Micrography

Figure 10 shows the typical micrography of the welded specimens. The three typical metallurgical zones are visible: base material (BM), fusion zone (FZ), and heat-affected zone (HAZ). After visual inspection, as demonstrated in Figure 10, the samples do not show cracks, which is in accordance with results seen in the literature [2,18,31,32]. It is observed that there are some spatters (see white arrows) on the additive and conventional sides (Figure 10a,b,f).

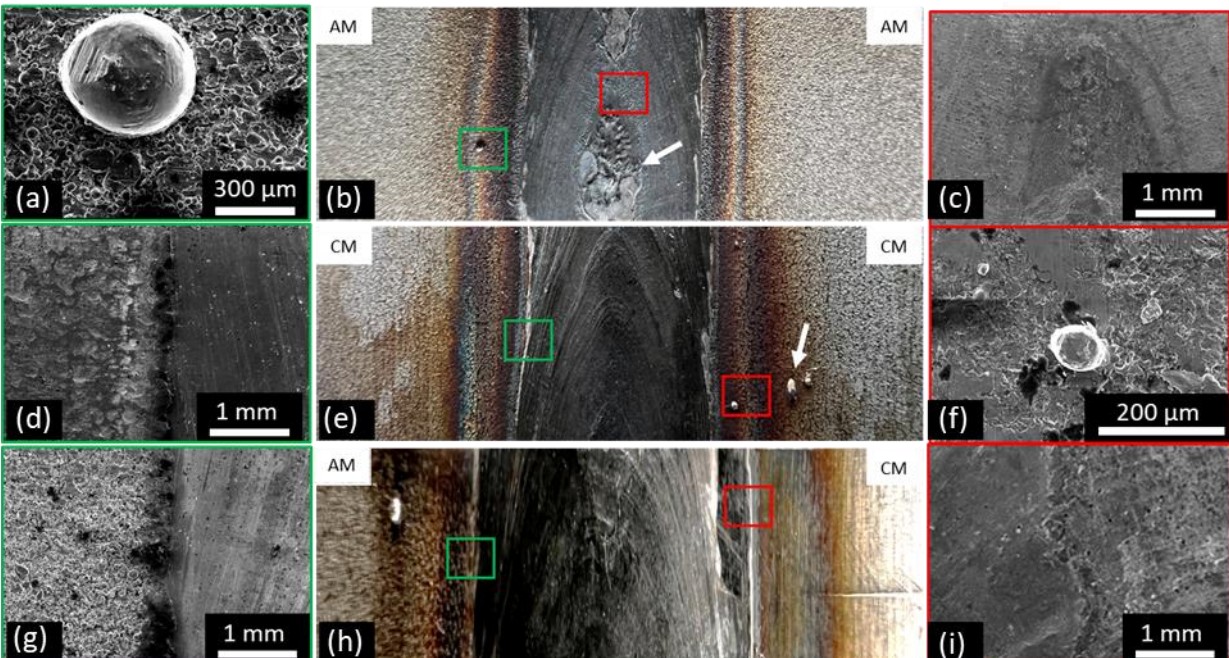

**Figure 10.** Welded joint macro and micrography: (**a**–**f**) spatters, (**b**) AM-AM, (**c**) AM-AM fusion zone, (**d**–**i**) HAZ CM side, (**e**) CM-CM, (**h**) CM-AM.

The granulated texture observed on the additive side is due to the last layer of powder that is not fully fused to the part during the L-PBF process (Figure 10a,g). Based on the observations, for both materials (AM and CM), the texture in the HAZ is similar to that of base material (BM). The grains morphology is characterized by an equiaxial granulometry in the BM and a directional grain growth in the FZ.

Visually, we can say that the welding method does not present any surface quality issues that could affect the mechanical properties in any of the configurations.

For welded configurations, cross-sections are observed with SEM. Figure 11 presents the results of these observations. The three metallurgical zones listed below are visible (Figure 11a–c). By observing the images of the two base materials (Figure 11g,h), we can see that both conventional and AM materials contain partial porosity. CM material exhibits a typical cold-rolled microstructure, where the direction of the rolling can be clearly seen (Figure 11h). It is also visible that the fusion zone is similar on both the AM and CM sides (Figure 11f,h,i). The latter contains some pores (Figure 11d,e). Some anomalies of material discontinuity are detected at the transition from the fused zone to the HAZ of the conventional side in the case of heterogonous welding (Figure 11f). The width of HAZ varies between 2 and 3 mm for all the configurations. The surface of the AM base material (Figure 11h) presents a typical micrography of the parts resulting from the L-PBF process (layer by layer). Similar images were presented in the study of Laitinen et al. [20] and in the technical data sheet of the EOS powder.

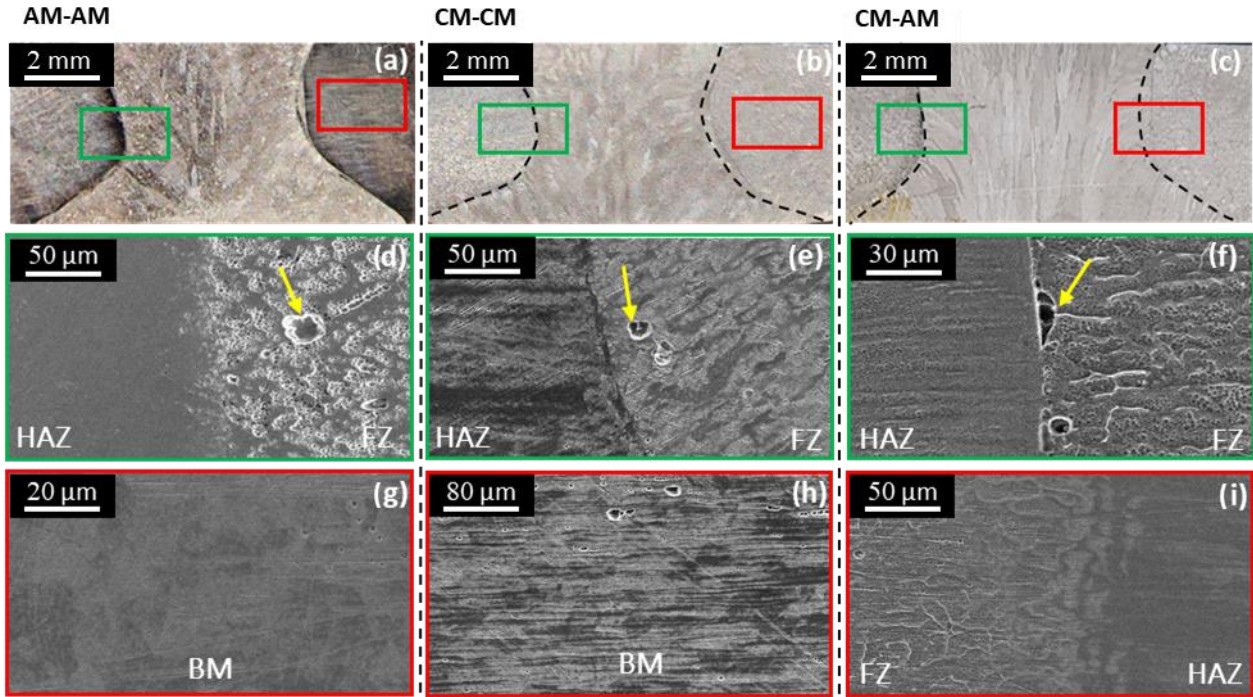

**Figure 11.** Microstructure of cross-section of welded samples: (**a**–**c**) higher magnification of welded cross sections, (**d**–**f**,**i**) fusion zone (FZ) and heat-affected zone (HAZ), (**g**,**h**) conventional and additive manufacturing base material.

### 3.2. Microhardness

Based on the microhardness measurements along the cross-section welded specimens shown in Figure 12, it is remarkable that welding decreases the microhardness values as it approaches the welded joint. Figure 12b,c show just half of the measurements because the welding is homogeneous, and the two half curves are similar. By looking at the average hardness value, the microhardness decrease is more significant for AM welded parts than for CM welded specimens: 56% (14 HRC to 6.2 HRC) compared to 40% (9.5 HRC to 5.6 HRC), (Figure 12b,c). For the AM-CM configuration, the tested specimen shows a smaller microhardness in the fusion zone than those of the two base materials. There is a difference of 7.5 HRC for the AM side compared to 1.1 HRC for the CM side. These microhardness variations are consistent with the results found in the literature [14,30].

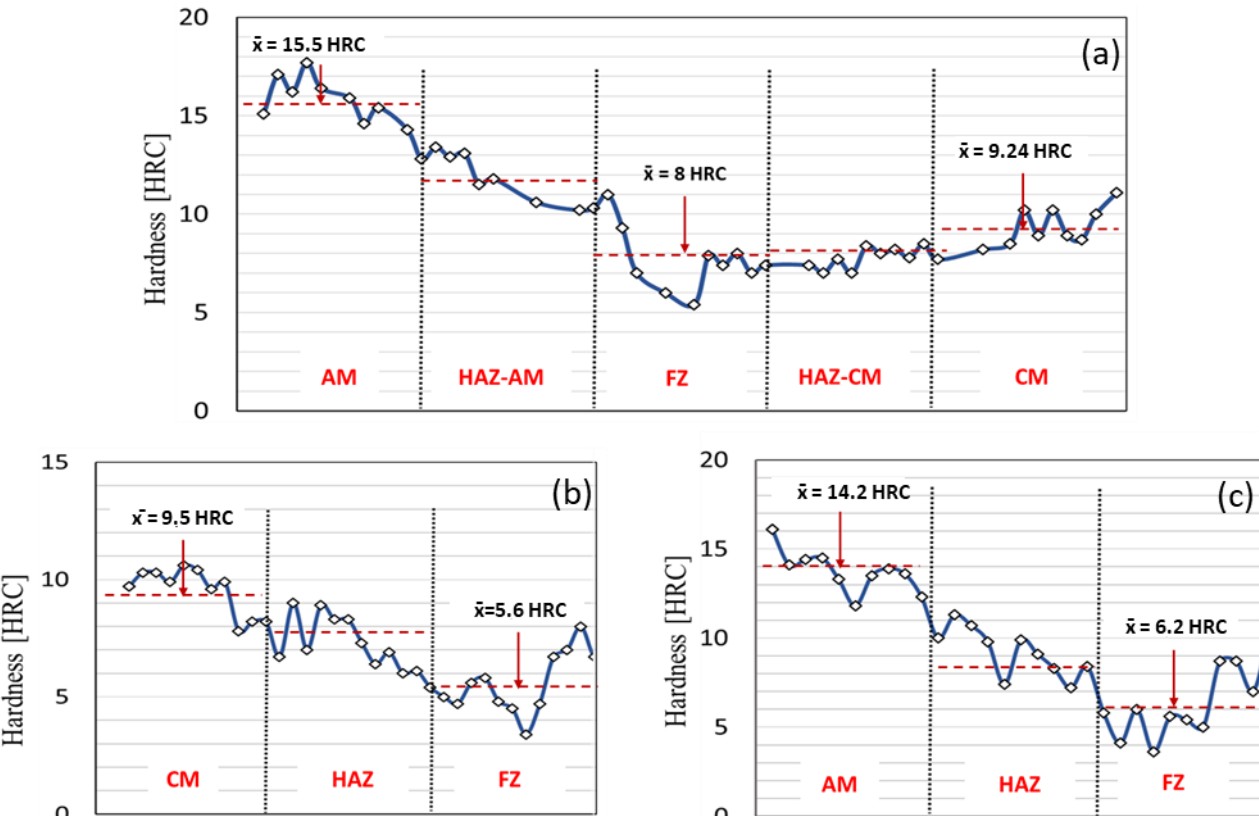

**Figure 12.** Microhardness profile (HRC) of cross-section welded samples, (**a**) CM-AM configuration, (**b**) CM-CM configuration, (**c**) AM-AM configuration.

*3.3. Tensile Test*

For each configuration, three tests are carried out. Average values of ultimate tensile strength, yield stress, and elongation at rupture are shown in Table 7 and Figure 13. Figure 14 presents one of the three tensile test curves for each configuration. The tested specimens show elastoplastic behavior with high ductility. The extensometer used has an elongation capacity of 20%. As a result, the values of elongation at break are deducted from the machine actuator displacement. Even if the values are not precise, this method provides a good approximation of elongation at break. As Table 7 and Figure 13 show, good reproducibility is observed. The non-welded AM specimen shows higher yield stress, lower ultimate tensile strength, and higher elongation at break than non-welded conventional specimens.

**Table 7.** Tensile test results of reference and welded samples.

|  | CM | AM | CM-CM | AM-AM | CM-AM |
|---|---|---|---|---|---|
| Yield stress (MPa) | $326 \pm 1$ | $414 \pm 3$ | - | - | - |
| Ultimate tensile strength (MPa) | $631 \pm 2$ | $629 \pm 2$ | $630 \pm 5$ | $622 \pm 2$ | $627 \pm 4$ |
| Elongation at break (%) | $55 \pm 4$ | $30 \pm 1$ | $40 \pm 2$ | $23 \pm 4$ | $29 \pm 2$ |
| Failure position | Base metal | | Welded material | | |

For all the values presented in the table, the tolerance intervals are equal to: $\pm (X_{max} - X_{min})/2$.

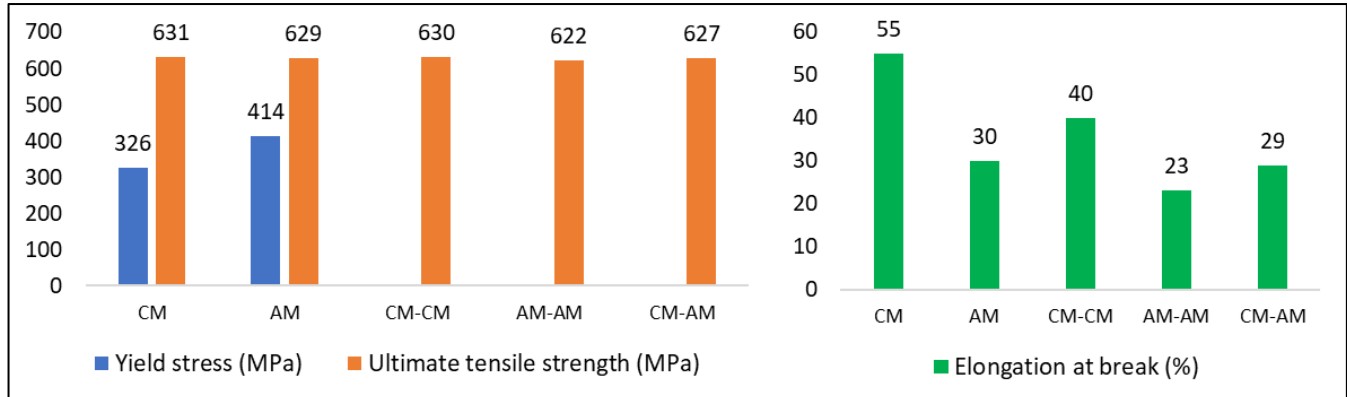

**Figure 13.** Tensile test results of reference and welded samples.

Compared to the non-welded specimens, all the welded configurations show a decrease in the elongation at break and yield stress. The ductility average of the welded configurations has established a 28% loss compared to the non-welded configurations. Regarding ultimate tensile strength, the results do not show a significant difference between non-welded and welded specimens.

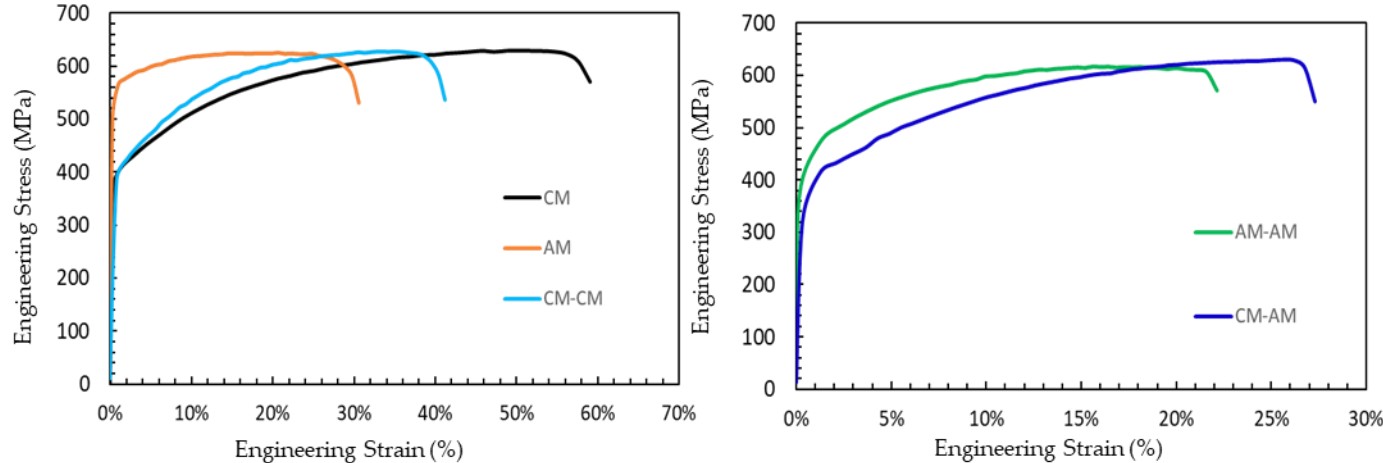

**Figure 14.** Stress–strain tensile curves of reference and welded samples.

Table 8 and Figure 15 show comparable values from the powder supplier EOS and from Mokhtari et al. [14]. For non-welded AM specimens, the results found in this study are lower than those mentioned in the EOS specifications, although the difference is very low (2%) for the ultimate stress. For convention material, except for the elongation at break, which is lower in our study, the results are comparable with Mower et al.'s study [28].

**Table 8.** Tensile test results in a similar study and supplier values for L-PBF material.

|  | AM, EOS | AM [14] | AM [28] | CM [14] | CM [28] | CM-CM [14] | AM-AM [14] |
|---|---|---|---|---|---|---|---|
| Yield stress (MPa) | 540 | 423 ± 5 | 496 | 261 ± 19 | 345 | - | - |
| Ultimate tensile strength (MPa) | 640 | 568 ± 5 | 717 | 602 ± 2 | 563 | 598 ± 3 | 568 ± 2 |
| Elongation at break (%) | 40 | 51 ± 1 | 28 | 73 ± 6 | 30 | 56 ± 3 | 42 ± 14 |

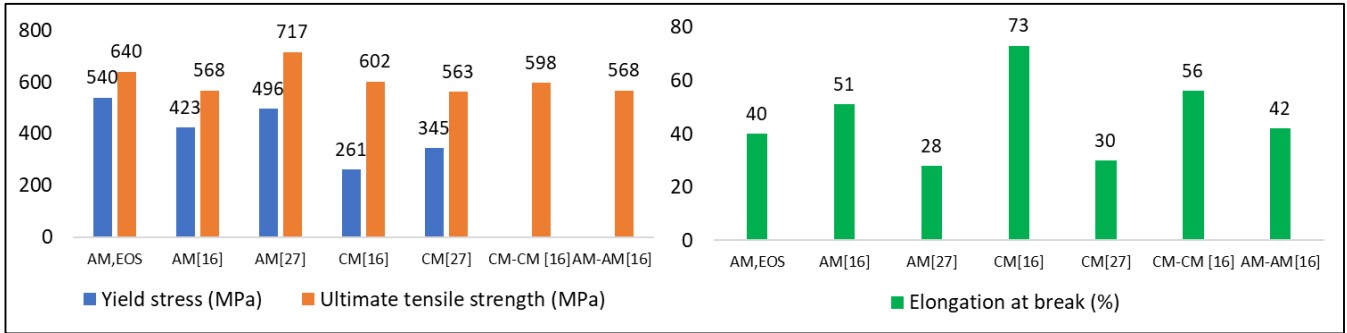

**Figure 15.** Tensile test results in a similar study and supplier values for L-PBF material.

For the welded configurations, the values are comparable and even partially better than those found in the study Mokhtari et al. [14], (see Table 8). In comparison to our study, Mokhtari et al. [14] did not use the same filler material and welding method, which may explain why our results differ from theirs.

The results of the tensile tests are consistent with those found in the microhardness tests. In the fused zone, the material is softer, which explains the lower ultimate limit after welding. It was also noticed that the values obtained are better than those of the filler material even if the rupture happened in the weld bead. In fact, in the fusion zone, a mixture of materials will be created with different proportions, which explains this increase [16]. In addition, the effect of rapid cooling during welding may be a factor.

### 3.4. Fatigue Test

For all configurations, the stress values retained for the fatigue tests are based on the ultimate tensile strength obtained from the tensile tests. The different load levels used in our fatigue tests are chosen as a percentage of the ultimate tensile strength. In order to minimize the effect of surface finish, especially since the specimens come from different manufacturing processes, manual polishing on the machined surfaces is performed on all specimens undergoing the fatigue test. As an indication, the roughness average measured with a roughness tester for CM-CM and AM-AM welded parts are shown in Table 9.

**Table 9.** Roughness measurement after polishing.

| Specimen | Ra1 (μm) | Ra2 (μm) | Ra3 (μm) | Mean |
|----------|----------|----------|----------|------|
| CM-CM | 0.75 | 0.64 | 0.67 | 0.69 |
| AM-AM | 0.79 | 0.61 | 0.77 | 0.72 |

Figure 16 shows the high-cycle fatigue regime (HCF) curves obtained for three load levels and three repetitions at each load. The fatigue resistance of non-welded specimens is higher than for those that are welded. Non-welded AM parts show slightly better fatigue performance than non-welded CM parts, despite the fact that CM and AM parts have almost the same ultimate tensile strength. Nonetheless, for the three welded configurations, the curves are close to each other. Although there are only three repetitions at each load, the curve of the heterogeneous welding (CM-AM) is framed by the two other configuration (CM-CM and AM-AM) curves and the welded CM specimens have the highest fatigue curve and resistance of the three welded configurations.

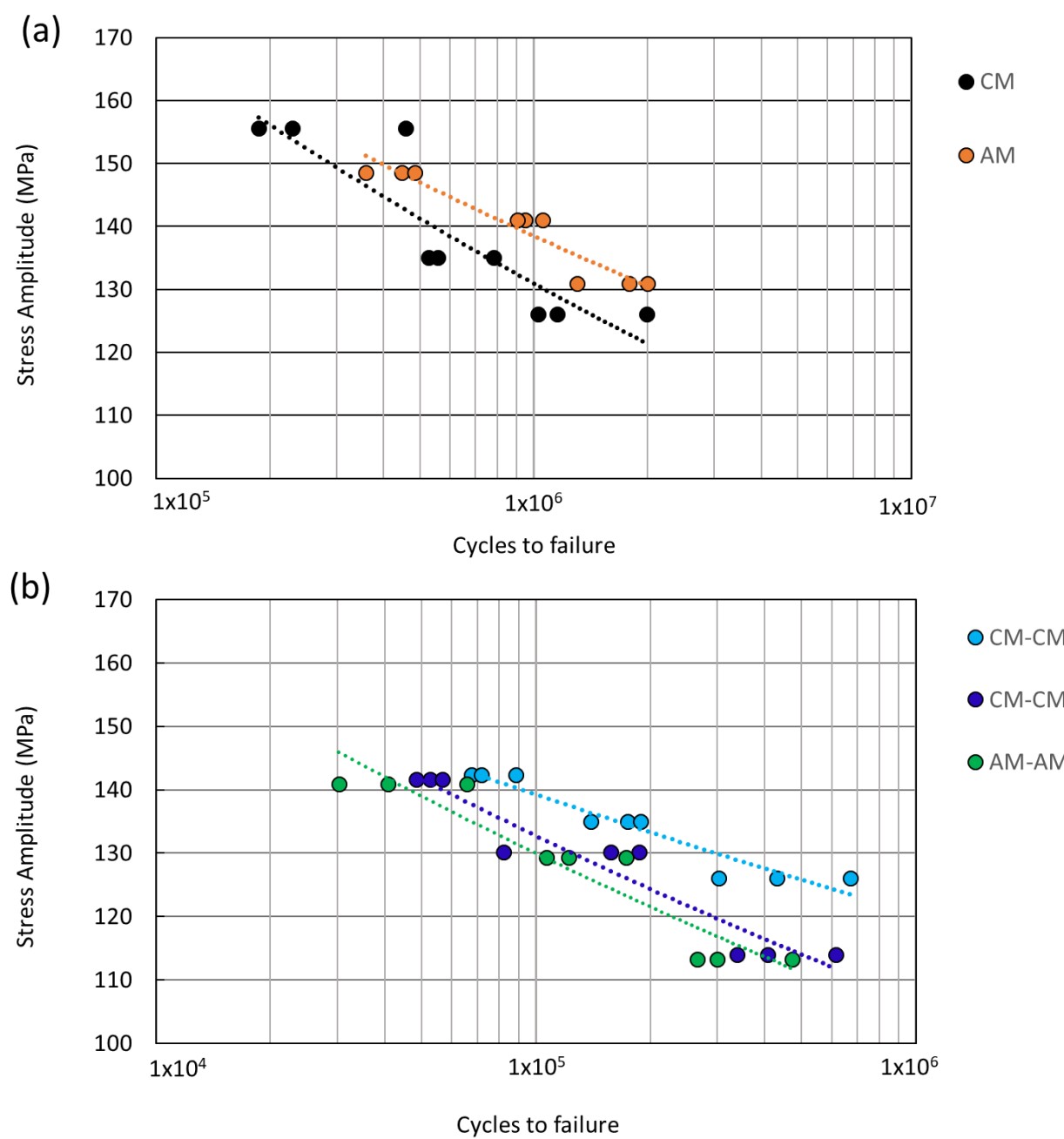

**Figure 16.** Fatigue finite life range: (**a**) Non-welded specimens (CM, AM), (**b**) welded parts (CM-CM, CM-AM, AM-AM).

As shown in Table 10, welded specimens have an approximately 30% lower endurance limit than the non-welded specimens. The values of the estimated endurance limit are consistent with those obtained in the finite life range. The order of the endurance limit values follows the position of the curves; the highest endurance limit value corresponds to the highest curve.

**Table 10.** Estimation of endurance limits.

| | CM | AM | CM-CM | CM-AM | AM-AM |
|---|---|---|---|---|---|
| Endurance limit estimation (MPa) | 122.14 | 125.92 | 90.44 | 88.40 | 83.85 |
| Standard deviation estimate | 5.30 | 8.56 | 5.30 | 5.30 | 30.27 |
| Validity condition | 0.19 | 0.50 | 0.22 | 0.25 | 1.84 |

In addition to residual stress caused by welding, the L-PBF process, as described in [33], also generates residual stress, mainly due to melting and solidification. Given that all the breaks occurred in the filler material, this aspect has been disregarded.

In general, and in the special case of this study, the mechanical properties of the filler metal are not exactly equal to those of the base metals. This has to be considered when interpreting fatigue results. All the welded specimens have a smaller endurance limit than those of the non-welded ones and all the fatigue failures are in the weld filler metal.

During the staircase test of the AM-AM specimens, one of the specimens fails at a non-expected load and with low number of cycles, which leads to a decrease in the endurance limit value and to a fifth load level. This low-load fatigue failure explains the lower endurance limit of the welded configuration AM-AM compared to the other two welded configurations and the value of 30.27 obtained for the standard deviation. The microscopic images of this fatigue failure are reported in Figure 17.

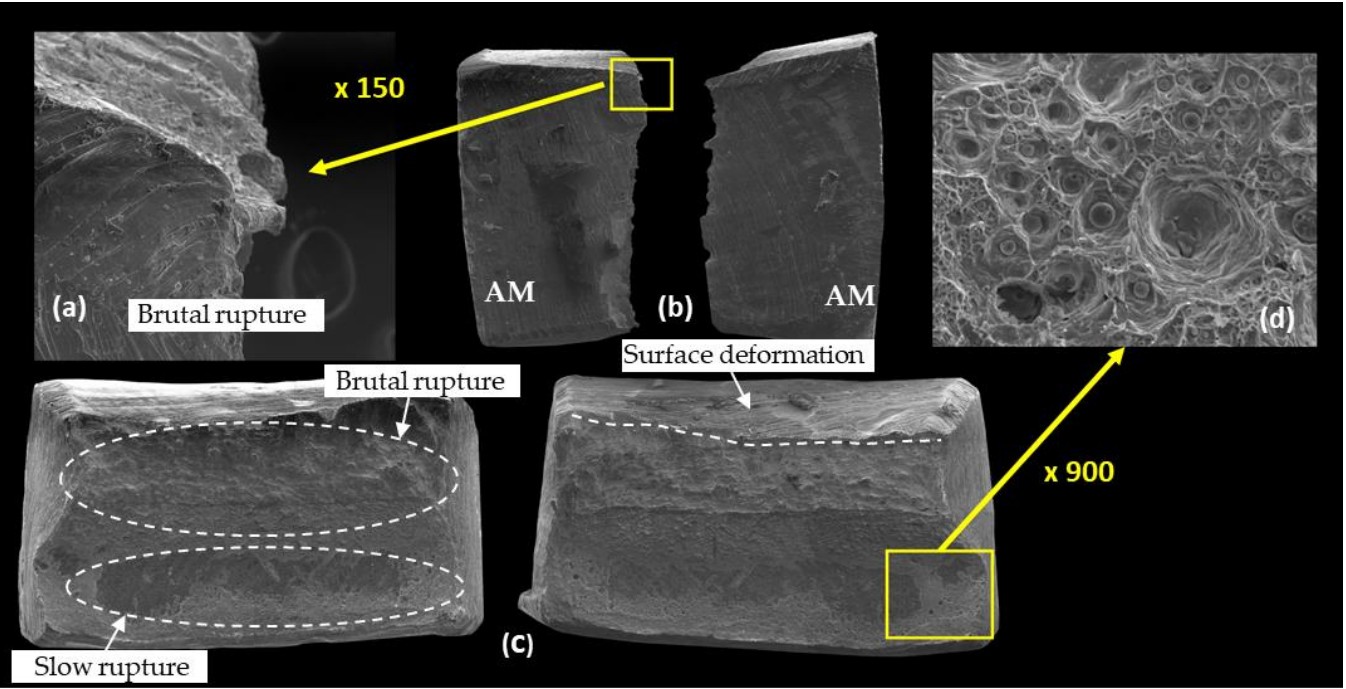

**Figure 17.** SEM images exhibiting the fracture surfaces of fatigue-tested AM-AM specimen: (**a**) macrography of the crack initiation, (**b**) fracture micrography on the welding surface, (**c**) failure surface macrography, (**d**) porosity in the failure surface.

The fracture surfaces show an area reduction (Figure 17). Two main areas are evident: the first presents the gradual crack propagation surface and the second shows the final and brutal fracture surface. The lack of density in the fractured area might justify the rapid rupture of this test specimen. Figure 17c shows that porosities are present in the fatigue area.

When compared visually with other observed fracture surfaces, this specimen has more pores (see Figure 18a,b). It is also remarkable that for all observed fracture surfaces, the surface anomalies are always in the lower part (root surface) of the specimen during

welding (Figure 18a,b). This observation could explain the density of the melt decreasing proportionally to the specimen depth.

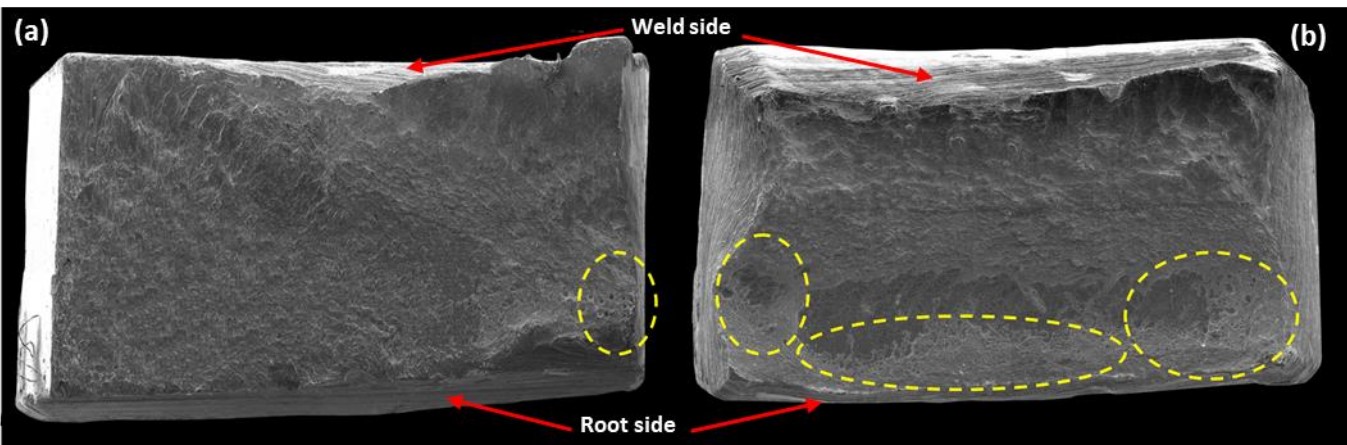

**Figure 18.** Failure surfaces at the staircase method: (**a**) AM-AM specimen fractured at (N: 1 757 738 cycle, σa: 87.85 MPa), (**b**) AM-AM specimen fractured at (N: 239 732 cycle, σa: 67.85 MPa).

For all samples, the observations of the obtained fractures show that the break is always in the middle of the sample (the welded area). However, the fractures are not similar and the distribution of the final fracture area and the crack propagation area varies from one specimen to another.

In their study, Mokhtari et al. [16] suggested that the porosity in the AM specimen welded by the L-PBF process can be transferred to the fusion zone, which can lead to this unexpected rupture. In the present study, the welding was performed manually using the GMAW-P process and the lack of density at the welding bead zone can be explained more easily by a welding defect. In fact, the welding path, the torch angle, and the proximity are variable all along the specimen's width. Therefore, we suggest that the unexpected rupture obtained was caused by a variation in weld quality.

## 4. Conclusions

This study evaluated the weldability of 316L stainless steel additive manufacturing parts through tensile and fatigue tests. Welding was performed manually using the GMAW-P process with a filler material EN ISO 14343-A g 18 8 MN with a ultimate tensile strength 5% lower than that of the 316L stainless steel tested in our research. All welded joints were of the butt joint type. Five configurations of specimens were studied: non-welded AM, AM with butt-welded joint, AM and laser cut (CM) with a butt-welded joint, laser cut (CM) with a butt-welded joint, and non-welded laser cut (CM). The tensile properties, high-cycle fatigue life curves, and endurance limits were obtained through testing according to the standards ASTM E8/E8M, ASTM E466-15, and ISO 12107: 2003. The fatigue tests were conducted at 20 Hz, with a stress ratio of R = 0.5. For each group of specimens, we performed three tests at three load levels, and we applied the staircase method to determine the endurance limit at two million cycles. The main conclusions of this study are as follows:

1. All static and fatigue fractures occurred in the filler material, notwithstanding the HAZ that could affect base material properties.
2. The tensile strengths of the different welded configurations were almost equal, with a maximum variation of 1.4%; this is due to the filler material having an ultimate limit almost equal to the 316L stainless steel.
3. In fatigue, the non-welded specimens (AM and CM) exhibited almost the same behavior. In the finite life domain, the difference was less than 10%. The endurance limit at two million cycles, as obtained with the staircase method, was practically equal.

4. The fatigue curves and endurance limits of the welded components were lower than for the two non-welded reference configurations. The two reference configurations give very comparable values with a variation of 3.7%.

5. For the AM-AM configuration, one unexpected failure occurred where a runout was expected. The endurance limit of this configuration was slightly lower and the standard deviation was substantially larger than those of the other welded configurations. The analyses of the fracture surface revealed that the failure was due to a welding defect.

Apart from the above unexpected failure, the results indicate that AM parts made of 316L stainless steel can be welded and that the strength of the butt-welded joint, in static and in fatigue, is comparable to that of the welds on specimens made by laser cutting.

This study demonstrates that AM parts made of 316L stainless steel can be welded like laser cut components. The results for 316L stainless steel show that 3D-printed parts can be welded with other printed parts or with parts produced by conventional processes with the same level of performance. Moreover, considering that printing chamber space is limited, it may be possible to divide a large part into several pieces, print the different pieces, and weld them together. The results also show that it is possible to repair additively manufactured parts through welding without a significant decrease in static and fatigue performance.

Future work may include other experimental designs by changing manufacturing parameters (welding method, material, additive manufacturing parameters, and heat treatment). In addition, the differences in fatigue failure initiation point warrant further investigation.

**Author Contributions:** Conceptualization, H.S. and J.B.; Methodology, H.S., J.B., G.C.-G., S.G. and J.D.; validation, J.B.; Investigation, H.S., J.B., C.B.; Writing—original draft preparation, H.S.; Writing—review and editing, J.B. and H.S.; Supervision, J.B. All authors have read and agreed to the published version of the manuscript.

**Funding:** This research was funded by NSERC, grant CDEPJ/507533.

**Data Availability Statement:** The data presented in this study are available in article.

**Acknowledgments:** We would like to thank Charles-André Fraser, Pedram Farhadipour, Suzie Loubert, and Richard Lafrance for their technical support.

**Conflicts of Interest:** The authors declare no conflict of interest.

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
