# Peer review of "Weldability of 316L Parts Produced by Metal Additive Manufacturing"

_jmmp, doi:10.3390/jmmp7020071_

Round 1

Reviewer 1 Report

Dear Authors, Your Paper needs MAJOR REVISION and I strongly suggest to incorporate the below mentioned changes in Your Paper and to send the revised paper back to me:-

1.      Introductory section does not give a clear picture about the need for carrying out welding in this research work. I STRONGLY ADVICE the authors to read the below mentioned research papers thoroughly to get idea about how to strengthen Your “Introduction” Part:

·         Satheesh C, Sevvel P and Senthil Kumar R: “Experimental Identification of Optimized Process Parameters for FSW of AZ91C Mg Alloy Using Quadratic Regression Models”, Strojniski Vestnik / Journal of Mechanical Engineering, Vol. 66 (12), 2020, pp. 736 – 751. https://doi.org/10.5545/sv-jme.2020.6929

Authors are also STRONGLY ADVICED TO CITE THE ABOVE PAPERS in the Introduction Section.

2.      In the Chapter 2, under the section 2.2, Authors are advised to include the photograph of the GMAW-P welding set up being employed in this experimental work.

Author Response

Response to Reviewer 1 Comments
Point 1: Introductory section does not give a clear picture about the need for carrying out welding in 
this research work. I STRONGLY ADVICE the authors to read the below mentioned research papers 
thoroughly to get idea about how to strengthen Your “Introduction” Part:
· Satheesh C, Sevvel P and Senthil Kumar R: “Experimental Identification of Optimized 
Process Parameters for FSW of AZ91C Mg Alloy Using Quadratic Regression Models”, Strojniski 
Vestnik / Journal of Mechanical Engineering, Vol. 66 (12), 2020, pp. 736 – 751. 
https://doi.org/10.5545/sv-jme.2020.6929 Authors are also STRONGLY ADVICED TO CITE THE 
ABOVE PAPERS in the Introduction Section.
Response 1: The proposed paper is interesting in terms of comparison method (experimental design), 
however, additive manufacturing is not the manufacturing process used in this research. The 
following research papers (already mentioned in the manuscript) address the subject of the need for 
carrying out welding : [13,14,15,16,17].
Point 2: In the Chapter 2, under the section 2.2, Authors are advised to include the photograph of the 
GMAW-P welding set up being employed in this experimental work.
Response 2: no other pictures of the GMAW-P set-up were taken during the preparation of the test 
specimens, however, figure 5 shows the mounting of the specimens before welding

Reviewer 2 Report

The paper analyzes the mechanical properties (tensile and fatigue) of GMAW welded joints made up from components of AISI 316L steel produced by AM (PBF technique) and conventional manufacturing.

The topic is of technological relevance since components obtained by AM are increasingly employed in industrial production and their performance after joining is crucial.

The manuscript is well structured and references are appropriate and up-to-date. English style is good. The experimental plan is clearly described and logically well structured.

The conclusions are consistent with the aim of the work.

In general it is an excellent paper that can be published as it is.

Just few minor observations can be made:

1) End of page 11, figure 12h instead of 12g should be addressed and the sentence should read "The surface of the AM base material (Figure 12, h) presents a typical micrography of the parts resulting from the L-PBF process (layer by layer)."

2) In Fig. 12 a, b and c the marker is missing and it could be useful to add information on the chemical etching used to reveal the microstructure and on the microscopy technique (optical or SEM)

Author Response

Response to Reviewer 2 Comments
Point 1: End of page 11, figure 12h instead of 12g should be addressed and the sentence should read 
"The surface of the AM base material (Figure 12, h) presents a typical micrography of the parts 
resulting from the L-PBF process (layer by layer)."
Response 1: Change performed
Point 2: In Fig. 12 a, b and c the marker is missing and it could be useful to add information on the 
chemical etching used to reveal the microstructure and on the microscopy technique (optical or SEM).
Response 2: Modification performed : Marker added

Reviewer 3 Report

Comments about the article “Weldability of 316L parts produced by metal additive manufacturing”

The authors presented an important contribution to the academic and industrial communities since a lack in the literature is the union of AM-ed parts, which must be studied and evaluated. The paper discusses the weldability of AM-ed 316L as AM-AM or AM-CM by the GMAW process, making a characterization of microstructures and hardness on the welded area. The material fatigue performance was proved, resulting in acceptably lower fatigue life for the welded samples compared to CM ones.

Some points could be improved or edited by the authors, as listed:

1. Introduction

2nd paragraph: The authors present only four strategies or techniques for metal AM “direct energy deposition, powder bed fusion, sheet lamination, and binder jetting [6]”. Please consider the Cold Spray deposition as a possibility. Some references to be listed:

doi.org/10.1016/j.addma.2018.04.017

doi.org/10.3390/coatings13020267

doi.org/10.1016/j.msea.2018.02.094

doi.org/10.3390/ma15196785

5th paragraph, Line 19: The authors of the ref [17] need to be corrected. The correct one is “Huysmans et al.” instead of “Humans et al.”

7th paragraph: The authors present the advantages of 316L and that this material has been used as powder for AM processing. The authors should list examples of using the material for some different process:

PBF – doi.org/10.1007/s40194-021-01098-z (ref already used by the authors)

DED – doi.org/10.1016/j.aime.2022.100079

WAAM – doi.org/10.1016/j.jmrt.2022.08.169

Cold Spray – doi.org/10.3390/coatings11020168

8th paragraph: The abbreviature GMAW-P is not defined previously in the text. The authors should add: “GMAW-P (Pulsed Gas Metal Arc Welding)”. It is in the Materials and Methods section but should be moved to the first apparition of the abbreviature.

2. Materials and Methods

Figures 3 and 4: The angles presented in Figure 3 (0, angled, and 90) are not seen in the AM-ed samples shown in Figure 4. The results and discussions section also does not present results considering the manufacturing angle. Therefore, the authors should eliminate Figure 3.

2.1. Test Piece Fabrication

Figure 2: there is a list of spectres (spectre 1, spectre 2, and spectre 3), but one only curve spectre is presented. The authors should eliminate the legend box.

2.2. Test piece welding

1st paragraph, Line 1: Use the past to describe the processes. Rewrite “Welding is performed…” as “Welding was performed…”

Table 5: use the same name for filler metal in the table and in the text. “EN ISO 14343-A-G 18 8 MN” or “307SI”

2nd paragraph, Line 1: The authors should add the manufacturer of the 307SI filler metal.

7th paragraph, Line 5: Do not use first person to describe the procedure. Eliminate the sentence “According to our partner, the used welding procedure …”, rewriting it as “The welding procedure met…”.

2.3. Angular Distortion Measurement

Figure 7: use point as a decimal sign, not coma. The authors should rewrite 1,57 and 1,58 as 1.57 and 1.58, respectively.

2.4. Tensile Test Protocol

1st paragraph, Line 6: insert a space between 0.0025 and mm/s.

1st paragraph, Line 6: keep 2000 and MPa in the same line.

Figure 8: use point as a decimal sign, not coma. The authors should rewrite 15,5, 9,5, and 14,2 as 15.5, 9.5, and 14.2, respectively.

2.5. Fatigue Test Protocol

Figure 9: use point as a decimal sign, not coma. The authors should rewrite 32,29 as 32.29.

2.6 Micrographic observation and microhardness test protocol

Figure 10: The authors should provide a scale bar in the image.

3. Results and Discussions

3.1. Micrography

The authors should comment here the grains morphology, the equiaxial granulometry in the BM and directional grain growth in the FZ.

1st paragraph, Line 3: “As demonstrated and in accordance with literature for 316L material welding, no cracks were found in the welded joints [2, 29, 30, 18, 19]”. It should be better written as “After visual inspection as demonstrated in 11 Figure, the samples did not represent cracks, which is in accordance with results seen in the literature [2, 29, 30, 18, 19]”.

4th paragraph, Linea 10: The authors should insert a space between 3 and mm.

3.2. Microhardness

Figure 13: use point as a decimal sign, not coma. The authors should rewrite 12,50 as 12.50.

3.3. Tensile Testing

The results could be presented in two graphs of bars (one for stress and one for elongation) beside de curves (Figure 16). It could compile all the information, making tables 7 and 8 obsolete.

1st paragraph, Line 2: Rewrite “figure 14” with a capital letter, “Figure 14”.

1st paragraph, Line 7: Rewrite “figure 14” with a capital letter, “Figure 14”.

3rd paragraph, Line 1: Rewrite “figure 15” with a capital letter, “Figure 15”.

Figure 14: The authors have to edit the stress unit. Type “MPa” instead of “Mpa”.

3.4 Fatigue testing

Figure 17: use point as a decimal sign, not coma. The authors should rewrite the Y axis values using point or eliminating the decimal values, which are unnecessary in a graph scale. It is well written in Figure 16.

Figure 19: use point as a decimal sign, not coma. The authors should rewrite 87,85 and 67,85 as 87.85 and 67.85, respectively.

7th paragraph, Line 1. Do not use first person to present and discuss the results. Rewrite “…we can see an area reduction…” as “it is noticed that there is an area reduction…”

Conclusions

6th paragraph, Line 1. Do not use first person to present the conclusions. Rewrite the sentence “…we had only one occurrence…” in third person.

References

18/32 (56%) are newer than 5 years old. It is a good ratio, but the authors should improve the number of recent publications on the theme, which should indicate an actual interest of the academic community in the theme studied in this article.

Author Response

Response to Reviewer 3 Comments
Point 1: 2nd paragraph: The authors present only four strategies or techniques for metal AM “direct 
energy deposition, powder bed fusion, sheet lamination, and binder jetting [6]”. Please consider the 
Cold Spray deposition as a possibility. Some references to be listed:
doi.org/10.1016/j.addma.2018.04.017
doi.org/10.3390/coatings13020267
doi.org/10.1016/j.msea.2018.02.094
doi.org/10.3390/ma15196785
Response 1: One of theses references was added , as suggested : 7
Point 2: 5th paragraph, Line 19: The authors of the ref [17] need to be corrected. The correct one is 
“Huysmans et al.” instead of “Humans et al.”
Response 2: Change performed
Point 3: 7th paragraph: The authors present the advantages of 316L and that this material has been 
used as powder for AM processing. The authors should list examples of using the material for some 
different process:
PBF – doi.org/10.1007/s40194-021-01098-z (ref already used by the authors)
DED – doi.org/10.1016/j.aime.2022.100079
WAAM – doi.org/10.1016/j.jmrt.2022.08.169
Cold Spray – doi.org/10.3390/coatings11020168
Response 3: Some of theses references was added as suggested : 26,27
Point 4: 8th paragraph: The abbreviature GMAW-P is not defined previously in the text. The authors 
should add: “GMAW-P (Pulsed Gas Metal Arc Welding)”. It is in the Materials and Methods section 
but should be moved to the first apparition of the abbreviature.
Response 4: Modification performed
2
Point 5: Figures 3 and 4: The angles presented in Figure 3 (0, angled, and 90) are not seen in the AMed samples shown in Figure 4. The results and discussions section also does not present results 
considering the manufacturing angle. Therefore, the authors should eliminate Figure 3.
Response 5: Figure 3 has been deleted
Point 6: 2.1. Test Piece Fabrication
Figure 2: there is a list of spectres (spectre 1, spectre 2, and spectre 3), but one only curve spectre is 
presented. The authors should eliminate the legend box.
Response 6: The figure legend now states that the three spectra are overlapping
Point 7: 2.2. Test piece welding
1st paragraph, Line 1: Use the past to describe the processes. Rewrite “Welding is performed…” as 
“Welding was performed…”
Response 7: Modification performed
Point 8: Table 5: use the same name for filler metal in the table and in the text. “EN ISO 14343-A-G 18 
8 MN” or “307SI”
Response 8: Modification performed
Point 9: 2nd paragraph, Line 1: The authors should add the manufacturer of the 307SI filler metal.
Response 9: Modification performed
Point 10: 7th paragraph, Line 5: Do not use first person to describe the procedure. Eliminate the 
sentence “According to our partner, the used welding procedure …”, rewriting it as “The welding 
procedure met…”.
Response 10: Modification performed
Point 11: 2.3. Angular Distortion Measurement
Figure 7: use point as a decimal sign, not coma. The authors should rewrite 1,57 and 1,58 as 1.57 and 
1.58, respectively.
Response 11: Modification performed
3
Point 12: 2.4. Tensile Test Protocol
1st paragraph, Line 6: insert a space between 0.0025 and mm/s.
1st paragraph, Line 6: keep 2000 and MPa in the same line.
Response 12: Modification performed
Point 13: Figure 8: use point as a decimal sign, not coma. The authors should rewrite 15,5, 9,5, and 
14,2 as 15.5, 9.5, and 14.2, respectively.
Response 13: Modification performed
Point 14: 2.5. Fatigue Test Protocol
Figure 9: use point as a decimal sign, not coma. The authors should rewrite 32,29 as 32.29.
Response 14: Modification performed
Point 15: 2.6 Micrographic observation and microhardness test protocol
Figure 10: The authors should provide a scale bar in the image.
Response 15: Scale added
Point 16: 3.1. Micrography
The authors should comment here the grains morphology, the equiaxial granulometry in the BM and 
directional grain growth in the FZ.
Response 16: Comment added
Point 17: 1st paragraph, Line 3: “As demonstrated and in accordance with literature for 316L material 
welding, no cracks were found in the welded joints [2, 29, 30, 18, 19]”. It should be better written as 
“After visual inspection as demonstrated in 11 Figure, the samples did not represent cracks, which is 
in accordance with results seen in the literature [2, 29, 30, 18, 19]”.
Response 17: Modification performed
Point 18: 4th paragraph, Linea 10: The authors should insert a space between 3 and mm.
Response 18: Modification performed
4
Point 19: 3.2. Microhardness
Figure 13: use point as a decimal sign, not coma. The authors should rewrite 12,50 as 12.50.
Response 19: Modification performed
Point 20: 3.3. Tensile Testing
The results could be presented in two graphs of bars (one for stress and one for elongation) beside de 
curves (Figure 16). It could compile all the information, making tables 7 and 8 obsolete.
Response 20: the numerical values displayed in the tables show the variability (standard deviation) 
found in the measurements. For this reason, we prefer to keep the bar graphs and the tables
Point 21: 1st paragraph, Line 2: Rewrite “figure 14” with a capital letter, “Figure 14”.
Response 21: Modification performed
Point 22: 1st paragraph, Line 7: Rewrite “figure 14” with a capital letter, “Figure 14”.
Response 22: Modification performed
Point 23: 3rd paragraph, Line 1: Rewrite “figure 15” with a capital letter, “Figure 15”.
Response 23: Modification performed
Point 24: Figure 14: The authors have to edit the stress unit. Type “MPa” instead of “Mpa”.
Response 24: Modification performed
Point 25: 3.4 Fatigue testing
Figure 17: use point as a decimal sign, not coma. The authors should rewrite the Y axis values using 
point or eliminating the decimal values, which are unnecessary in a graph scale. It is well written in 
Figure 16.
Response 25: Modification performed
Point 26: Figure 19: use point as a decimal sign, not coma. The authors should rewrite 87,85 and 67,85 
as 87.85 and 67.85, respectively.
Response 26: Modification performed
5
Point 27: 7th paragraph, Line 1. Do not use first person to present and discuss the results. Rewrite 
“…we can see an area reduction…” as “it is noticed that there is an area reduction…”
Response 27: Modification performed
Point 28: Conclusions
6th paragraph, Line 1. Do not use first person to present the conclusions. Rewrite the sentence “…we 
had only one occurrence…” in
Response 28: Modification performed
Point 29: 18/32 (56%) are newer than 5 years old. It is a good ratio, but the authors should improve 
the number of recent publications on the theme, which should indicate an actual interest of the 
academic community in the theme studied in this article.
Response 29: The following references were added: 7,26,27

Round 2

Reviewer 1 Report

Dear Authors, I recommend the revised paper for publication

Reviewer 3 Report

The authors improved the work quality by accepting the reviewer's suggestions.